# Do Transformers Really Perform Bad for Graph Representation?

**Chengxuan Ying**[1]*, **Tianle Cai**[2], **Shengjie Luo**[3]*,
**Shuxin Zheng**[4]†, **Guolin Ke**[4], **Di He**[4]†, **Yanming Shen**[1], **Tie-Yan Liu**[4]
[1]Dalian University of Technology    [2]Princeton University
[3]Peking University    [4]Microsoft Research Asia
yingchengsyuan@gmail.com, tianle.cai@princeton.edu, luosj@stu.pku.edu.cn
{shuz†, guoke, dihe†, tyliu}@microsoft.com, shen@dlut.edu.cn

## Abstract

The Transformer architecture has become a dominant choice in many domains, such as natural language processing and computer vision. Yet, it has not achieved competitive performance on popular leaderboards of graph-level prediction compared to mainstream GNN variants. Therefore, it remains a mystery how Transformers could perform well for graph representation learning. In this paper, we solve this mystery by presenting Graphormer, which is built upon the standard Transformer architecture, and could attain excellent results on a broad range of graph representation learning tasks, especially on the recent OGB Large-Scale Challenge. Our key insight to utilizing Transformer in the graph is the necessity of effectively encoding the structural information of a graph into the model. To this end, we propose several simple yet effective structural encoding methods to help Graphormer better model graph-structured data. Besides, we mathematically characterize the expressive power of Graphormer and exhibit that with our ways of encoding the structural information of graphs, many popular GNN variants could be covered as the special cases of Graphormer. The code and models of Graphormer will be made publicly available at `https://github.com/Microsoft/Graphormer`.

## 1    Introduction

The Transformer [46] is well acknowledged as the most powerful neural network in modelling sequential data, such as natural language [11, 35, 6] and speech [17]. Model variants built upon Transformer have also been shown great performance in computer vision [12, 36] and programming language [19, 57, 41]. However, to the best of our knowledge, Transformer has still not been the de-facto standard on public graph representation leaderboards [22, 14, 21]. There are many attempts of leveraging Transformer into the graph domain, but the only effective way is replacing some key modules (e.g., feature aggregation) in classic GNN variants by the softmax attention [47, 7, 23, 48, 56, 43, 13]. Therefore, it is still an open question whether Transformer architecture is suitable to model graphs and how to make it work in graph representation learning.

In this paper, we give an affirmative answer by developing Graphormer, which is directly built upon the standard Transformer, and achieves state-of-the-art performance on a wide range of graph-level prediction tasks, including the very recent Open Graph Benchmark Large-Scale Challenge (OGB-LSC) [21], and several popular leaderboards (e.g., OGB [22], Benchmarking-GNN [14]). The Transformer is originally designed for sequence modeling. To utilize its power in graphs, we believe

---

*Interns at MSRA.
†Corresponding authors.

35th Conference on Neural Information Processing Systems (NeurIPS 2021).

the key is to properly incorporate structural information of graphs into the model. Note that for each node $i$, the self-attention only calculates the semantic similarity between $i$ and other nodes, without considering the structural information of a graph reflected on the nodes and the relation between node pairs. Graphormer incorporates several effective structural encoding methods to leverage such information, which are described below.

First, we propose a *Centrality Encoding* in Graphormer to capture the node importance in the graph. In a graph, different nodes may have different importance, e.g., celebrities are considered to be more influential than the majority of web users in a social network. However, such information isn't reflected in the self-attention module as it calculates the similarities mainly using the node semantic features. To address the problem, we propose to encode the node centrality in Graphormer. In particular, we leverage the *degree centrality* for the centrality encoding, where a learnable vector is assigned to each node according to its degree and added to the node features in the input layer. Empirical studies show that simple centrality encoding is effective for Transformer in modeling the graph data.

Second, we propose a novel *Spatial Encoding* in Graphormer to capture the structural relation between nodes. One notable geometrical property that distinguishes graph-structured data from other structured data, e.g., language, images, is that there does not exist a canonical grid to embed the graph. In fact, nodes can only lie in a non-Euclidean space and are linked by edges. To model such structural information, for each node pair, we assign a learnable embedding based on their spatial relation. Multiple measurements in the literature could be leveraged for modeling spatial relations. For a general purpose, we use the distance of the shortest path between any two nodes as a demonstration, which will be encoded as a bias term in the softmax attention and help the model accurately capture the spatial dependency in a graph. In addition, sometimes there is additional spatial information contained in edge features, such as the type of bond between two atoms in a molecular graph. We design a new edge encoding method to further take such signal into the Transformer layers. To be concrete, for each node pair, we compute an average of dot-products of the edge features and learnable embeddings along the shortest path, then use it in the attention module. Equipped with these encodings, Graphormer could better model the relationship for node pairs and represent the graph.

By using the proposed encodings above, we further mathematically show that Graphormer has strong expressiveness as many popular GNN variants are just its special cases. The great capacity of the model leads to state-of-the-art performance on a wide range of tasks in practice. On the large-scale quantum chemistry regression dataset[3] in the very recent Open Graph Benchmark Large-Scale Challenge (OGB-LSC) [21], Graphormer outperforms most mainstream GNN variants by more than 10% points in terms of the relative error. On other popular leaderboards of graph representation learning (e.g., MolHIV, MolPCBA, ZINC) [22, 14], Graphormer also surpasses the previous best results, demonstrating the potential and adaptability of the Transformer architecture.

## 2 Preliminary

In this section, we recap the preliminaries in Graph Neural Networks and Transformer.

**Graph Neural Network (GNN).** Let $G = (V, E)$ denote a graph where $V = \{v_1, v_2, \cdots, v_n\}$, $n = |V|$ is the number of nodes. Let the feature vector of node $v_i$ be $x_i$. GNNs aim to learn representation of nodes and graphs. Typically, modern GNNs follow a learning schema that iteratively updates the representation of a node by aggregating representations of its first or higher-order neighbors. We denote $h_i^{(l)}$ as the representation of $v_i$ at the $l$-th layer and define $h_i^{(0)} = x_i$. The $l$-th iteration of aggregation could be characterized by AGGREGATE-COMBINE step as

$$a_i^{(l)} = \text{AGGREGATE}^{(l)} \left( \left\{ h_j^{(l-1)} : j \in \mathcal{N}(v_i) \right\} \right), \quad h_i^{(l)} = \text{COMBINE}^{(l)} \left( h_i^{(l-1)}, a_i^{(l)} \right), \quad (1)$$

where $\mathcal{N}(v_i)$ is the set of first or higher-order neighbors of $v_i$. The AGGREGATE function is used to gather the information from neighbors. Common aggregation functions include MEAN, MAX, SUM, which are used in different architectures of GNNs [26, 18, 47, 50]. The goal of COMBINE function is to fuse the information from neighbors into the node representation.

---

[3] `https://ogb.stanford.edu/kddcup2021/pcqm4m/`

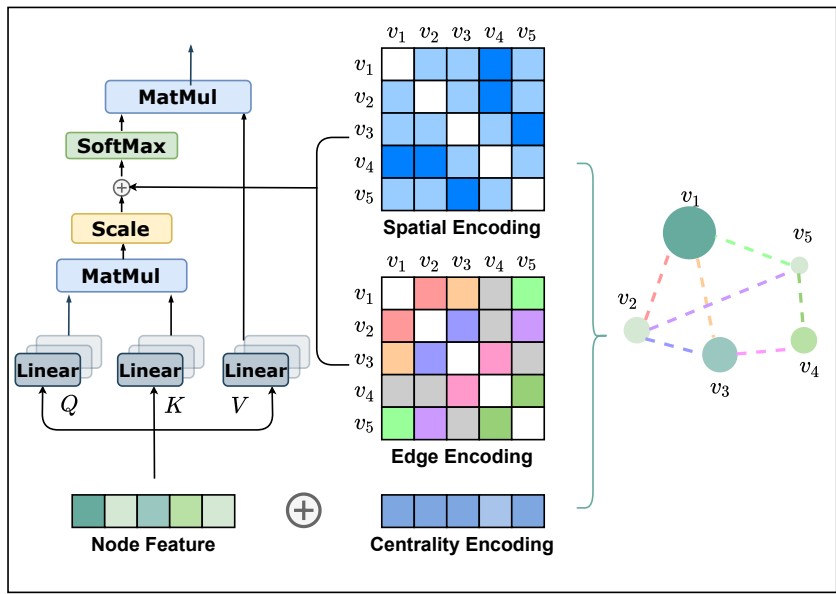

Figure 1: An illustration of our proposed centrality encoding, spatial encoding, and edge encoding in Graphormer.

In addition, for graph representation tasks, a READOUT function is designed to aggregate node features $h_i^{(L)}$ of the final iteration into the representation $h_G$ of the entire graph $G$:

$$h_G = \text{READOUT}\left(\left\{h_i^{(L)} \mid v_i \in G\right\}\right). \tag{2}$$

READOUT can be implemented by a simple permutation invariant function such as summation [50] or a more sophisticated graph-level pooling function [1].

**Transformer.** The Transformer architecture consists of a composition of Transformer layers [46]. Each Transformer layer has two parts: a self-attention module and a position-wise feed-forward network (FFN). Let $H = \left[h_1^\top, \cdots, h_n^\top\right]^\top \in \mathbb{R}^{n \times d}$ denote the input of self-attention module where $d$ is the hidden dimension and $h_i \in \mathbb{R}^{1 \times d}$ is the hidden representation at position $i$. The input $H$ is projected by three matrices $W_Q \in \mathbb{R}^{d \times d_K}, W_K \in \mathbb{R}^{d \times d_K}$ and $W_V \in \mathbb{R}^{d \times d_V}$ to the corresponding representations $Q, K, V$. The self-attention is then calculated as:

$$Q = HW_Q, \quad K = HW_K, \quad V = HW_V, \tag{3}$$

$$A = \frac{QK^\top}{\sqrt{d_K}}, \quad \text{Attn}(H) = \text{softmax}(A)V, \tag{4}$$

where $A$ is a matrix capturing the similarity between queries and keys. For simplicity of illustration, we consider the single-head self-attention and assume $d_K = d_V = d$. The extension to the multi-head attention is standard and straightforward, and we omit bias terms for simplicity.

# 3 Graphormer

In this section, we present our Graphormer for graph tasks. First, we elaborate on several key designs in the Graphormer, which serve as an inductive bias in the neural network to learn the graph representation. We further provide the detailed implementations of Graphormer. Finally, we show that our proposed Graphormer is more powerful since popular GNN models [26, 50, 18] are its special cases.

### 3.1 Structural Encodings in Graphormer

As discussed in the introduction, it is important to develop ways to leverage the structural information of graphs into the Transformer model. To this end, we present three simple but effective designs of encoding in Graphormer. See Figure 1 for an illustration.

#### 3.1.1 Centrality Encoding

In Eq.4, the attention distribution is calculated based on the semantic correlation between nodes. However, node centrality, which measures how important a node is in the graph, is usually a strong signal for graph understanding. For example, celebrities who have a huge number of followers are important factors in predicting the trend of a social network [38, 37]. Such information is neglected in the current attention calculation, and we believe it should be a valuable signal for Transformer models.

In Graphormer, we use the degree centrality, which is one of the standard centrality measures in literature, as an additional signal to the neural network. To be specific, we develop a *Centrality Encoding* which assigns each node two real-valued embedding vectors according to its indegree and outdegree. As the centrality encoding is applied to each node, we simply add it to the node features as the input.

$$h_i^{(0)} = x_i + z_{\deg^-(v_i)}^- + z_{\deg^+(v_i)}^+,\tag{5}$$

where $z^-, z^+ \in \mathbb{R}^d$ are learnable embedding vectors specified by the indegree $\deg^-(v_i)$ and outdegree $\deg^+(v_i)$ respectively. For undirected graphs, $\deg^-(v_i)$ and $\deg^+(v_i)$ could be unified to $\deg(v_i)$. By using the centrality encoding in the input, the softmax attention can catch the node importance signal in the queries and the keys. Therefore the model can capture both the semantic correlation and the node importance in the attention mechanism.

#### 3.1.2 Spatial Encoding

An advantage of Transformer is its global receptive field. In each Transformer layer, each token can attend to the information at any position and then process its representation. But this operation has a byproduct problem that the model has to explicitly specify different positions or encode the positional dependency (such as locality) in the layers. For sequential data, one can either give each position an embedding (i.e., absolute positional encoding [46]) as the input or encode the relative distance of any two positions (i.e., relative positional encoding [42, 44]) in the Transformer layer.

However, for graphs, nodes are not arranged as a sequence. They can lie in a multi-dimensional spatial space and are linked by edges. To encode the structural information of a graph in the model, we propose a novel *Spatial Encoding*. Concretely, for any graph $G$, we consider a function $\phi(v_i, v_j) : V \times V \to \mathbb{R}$ which measures the spatial relation between $v_i$ and $v_j$ in graph $G$. The function $\phi$ can be defined by the connectivity between the nodes in the graph. In this paper, we choose $\phi(v_i, v_j)$ to be the distance of the shortest path (SPD) between $v_i$ and $v_j$ if the two nodes are connected. If not, we set the output of $\phi$ to be a special value, i.e., -1. We assign each (feasible) output value a learnable scalar which will serve as a bias term in the self-attention module. Denote $A_{ij}$ as the $(i, j)$-element of the Query-Key product matrix $A$, we have:

$$A_{ij} = \frac{(h_i W_Q)(h_j W_K)^T}{\sqrt{d}} + b_{\phi(v_i, v_j)},\tag{6}$$

where $b_{\phi(v_i, v_j)}$ is a learnable scalar indexed by $\phi(v_i, v_j)$, and shared across all layers.

Here we discuss several benefits of our proposed method. First, compared to conventional GNNs described in Section 2, where the receptive field is restricted to the neighbors, we can see that in Eq. (6), the Transformer layer provides a global information that each node can attend to all other nodes in the graph. Second, by using $b_{\phi(v_i, v_j)}$, each node in a single Transformer layer can adaptively attend to all other nodes according to the graph structural information. For example, if $b_{\phi(v_i, v_j)}$ is learned to be a decreasing function with respect to $\phi(v_i, v_j)$, for each node, the model will likely pay more attention to the nodes near it and pay less attention to the nodes far away from it.

### 3.1.3 Edge Encoding in the Attention

In many graph tasks, edges also have structural features, e.g., in a molecular graph, atom pairs may have features describing the type of bond between them. Such features are important to the graph representation, and encoding them together with node features into the network is essential. There are mainly two edge encoding methods used in previous works. In the first method, the edge features are added to the associated nodes' features [22, 30]. In the second method, for each node, its associated edges' features will be used together with the node features in the aggregation [15, 50, 26]. However, such ways of using edge feature only propagate the edge information to its associated nodes, which may not be an effective way to leverage edge information in representation of the whole graph.

To better encode edge features into attention layers, we propose a new edge encoding method in Graphormer. The attention mechanism needs to estimate correlations for each node pair $(v_i, v_j)$, and we believe the edges connecting them should be considered in the correlation as in [34, 48]. For each ordered node pair $(v_i, v_j)$, we find (one of) the shortest path $\text{SP}_{ij} = (e_1, e_2, ..., e_N)$ from $v_i$ to $v_j$, and compute an average of the dot-products of the edge feature and a learnable embedding along the path. The proposed edge encoding incorporates edge features via a bias term to the attention module. Concretely, we modify the $(i, j)$-element of $A$ in Eq. (3) further with the edge encoding $c_{ij}$ as:

$$A_{ij} = \frac{(h_i W_Q)(h_j W_K)^T}{\sqrt{d}} + b_{\phi(v_i, v_j)} + c_{ij}, \text{ where } c_{ij} = \frac{1}{N} \sum_{n=1}^{N} x_{e_n} (w_n^E)^T, \qquad (7)$$

where $x_{e_n}$ is the feature of the $n$-th edge $e_n$ in $\text{SP}_{ij}$, $w_n^E \in \mathbb{R}^{d_E}$ is the $n$-th weight embedding, and $d_E$ is the dimensionality of edge feature.

## 3.2 Implementation Details of Graphormer

**Graphormer Layer.** Graphormer is built upon the original implementation of classic Transformer encoder described in [46]. In addition, we apply the layer normalization (LN) before the multi-head self-attention (MHA) and the feed-forward blocks (FFN) instead of after [49]. This modification has been unanimously adopted by all current Transformer implementations because it leads to more effective optimization [40]. Especially, for FFN sub-layer, we set the dimensionality of input, output, and the inner-layer to the same dimension with $d$. We formally characterize the Graphormer layer as below:

$$h'^{(l)} = \text{MHA}(\text{LN}(h^{(l-1)})) + h^{(l-1)} \qquad (8)$$
$$h^{(l)} = \text{FFN}(\text{LN}(h'^{(l)})) + h'^{(l)} \qquad (9)$$

**Special Node.** As stated in the previous section, various graph pooling functions are proposed to represent the graph embedding. Inspired by [15], in Graphormer, we add a special node called [VNode] to the graph, and make connection between [VNode] and each node individually. In the AGGREGATE-COMBINE step, the representation of [VNode] has been updated as normal nodes in graph, and the representation of the entire graph $h_G$ would be the node feature of [VNode] in the final layer. In the BERT model [11, 35], there is a similar token, i.e., [CLS], which is a special token attached at the beginning of each sequence, to represent the sequence-level feature on downstream tasks. While the [VNode] is connected to all other nodes in graph, which means the distance of the shortest path is 1 for any $\phi([\text{VNode}], v_j)$ and $\phi(v_i, [\text{VNode}])$, the connection is not physical. To distinguish the connection of physical and virtual, inspired by [25], we reset all spatial encodings for $b_{\phi([\text{VNode}], v_j)}$ and $b_{\phi(v_i, [\text{VNode}])}$ to a distinct learnable scalar.

## 3.3 How Powerful is Graphormer?

In the previous subsections, we introduce three structural encodings and the architecture of Graphormer. Then a natural question is: *Do these modifications make Graphormer more powerful than other GNN variants?* In this subsection, we first give an affirmative answer by showing that Graphormer can represent the AGGREGATE and COMBINE steps in popular GNN models:

**Fact 1.** *By choosing proper weights and distance function $\phi$, the Graphormer layer can represent AGGREGATE and COMBINE steps of popular GNN models such as GIN, GCN, GraphSAGE.*

The proof sketch to derive this result is: 1) Spatial encoding enables self-attention module to distinguish neighbor set $\mathcal{N}(v_i)$ of node $v_i$ so that the softmax function can calculate mean statistics over $\mathcal{N}(v_i)$; 2) Knowing the degree of a node, mean over neighbors can be translated to sum over neighbors; 3) With multiple heads and FFN, representations of $v_i$ and $\mathcal{N}(v_i)$ can be processed separately and combined together later. We defer the proof of this fact to Appendix A.

Moreover, we show further that by using our spatial encoding, Graphormer can go beyond classic message passing GNNs whose expressive power is no more than the 1-Weisfeiler-Lehman (WL) test. We give a concrete example in Appendix A to show how Graphormer helps distinguish graphs that the 1-WL test fails to.

**Connection between Self-attention and Virtual Node.** Besides the superior expressiveness than popular GNNs, we also find an interesting connection between using self-attention and the virtual node heuristic [15, 31, 24, 22]. As shown in the leaderboard of OGB [22], the virtual node trick, which augments graphs with additional supernodes that are connected to all nodes in the original graphs, can significantly improve the performance of existing GNNs. Conceptually, the benefit of the virtual node is that it can aggregate the information of the *whole graph* (like the READOUT function) and then propagate it to *each node*. However, a naive addition of a supernode to a graph can potentially lead to inadvertent over-smoothing of information propagation [24]. We instead find that such a graph-level aggregation and propagation operation can be naturally fulfilled by vanilla self-attention without additional encodings. Concretely, we can prove the following fact:

**Fact 2.** *By choosing proper weights, every node representation of the output of a Graphormer layer without additional encodings can represent MEAN READOUT functions.*

This fact takes the advantage of self-attention that each node can attend to all other nodes. Thus it can simulate graph-level READOUT operation to aggregate information from the whole graph. Besides the theoretical justification, we empirically find that Graphormer does not encounter the problem of over-smoothing, which makes the improvement scalable. The fact also inspires us to introduce a special node for graph readout (see the previous subsection).

# 4  Experiments

We first conduct experiments on the recent OGB-LSC [21] quantum chemistry regression (i.e., PCQM4M-LSC) challenge, which is currently the biggest graph-level prediction dataset and contains more than 3.8M graphs in total. Then, we report the results on the other three popular tasks: ogbg-molhiv, ogbg-molpcba and ZINC, which come from the OGB [22] and benchmarking-GNN [14] leaderboards. Finally, we ablate the important design elements of Graphormer. A detailed description of datasets and training strategies could be found in Appendix B.

## 4.1  OGB Large-Scale Challenge

**Baselines.** We benchmark the proposed Graphormer with GCN [26] and GIN [50], and their variants with virtual node (-VN) [15]. They achieve the state-of-the-art valid and test mean absolute error (MAE) on the official leaderboard[4] [21]. In addition, we compare to GIN's multi-hop variant [5], and 12-layer deep graph network DeeperGCN [30], which also show promising performance on other leaderboards. We further compare our Graphormer with the recent Transformer-based graph model GT [13].

**Settings.** We primarily report results on two model sizes: **Graphormer** ($L = 12, d = 768$), and a smaller one **Graphormer$_{\text{SMALL}}$** ($L = 6, d = 512$). Both the number of attention heads in the attention module and the dimensionality of edge features $d_E$ are set to 32. We use AdamW as the optimizer, and set the hyper-parameter $\epsilon$ to 1e-8 and $(\beta1, \beta2)$ to (0.99,0.999). The peak learning rate is set to 2e-4 (3e-4 for **Graphormer$_{\text{SMALL}}$**) with a 60k-step warm-up stage followed by a linear decay learning rate scheduler. The total training steps are 1M. The batch size is set to 1024. All models are trained on 8 NVIDIA V100 GPUS for about 2 days.

---

[4]`https://github.com/snap-stanford/ogb/tree/master/examples/lsc/pcqm4m#performance`

Table 1: Results on PCQM4M-LSC. * indicates the results are cited from the official leaderboard [21].

| method | #param. | train MAE | validate MAE |
|---|---|---|---|
| GCN [26] | 2.0M | 0.1318 | 0.1691 (0.1684*) |
| GIN [50] | 3.8M | 0.1203 | 0.1537 (0.1536*) |
| GCN-VN [26, 15] | 4.9M | 0.1225 | 0.1485 (0.1510*) |
| GIN-VN [50, 15] | 6.7M | 0.1150 | 0.1395 (0.1396*) |
| GINE-VN [5, 15] | 13.2M | 0.1248 | 0.1430 |
| DeeperGCN-VN [30, 15] | 25.5M | 0.1059 | 0.1398 |
| GT [13] | 0.6M | 0.0944 | 0.1400 |
| GT-Wide [13] | 83.2M | 0.0955 | 0.1408 |
| Graphormer$_{\text{SMALL}}$ | 12.5M | 0.0778 | 0.1264 |
| Graphormer | 47.1M | **0.0582** | **0.1234** |

**Results.**    Table 1 summarizes performance comparisons on PCQM4M-LSC dataset. From the table, GIN-VN achieves the previous state-of-the-art validate MAE of 0.1395. The original implementation of GT [13] employs a hidden dimension of 64 to reduce the total number of parameters. For a fair comparison, we also report the result by enlarging the hidden dimension to 768, denoted by GT-Wide, which leads to a total number of parameters of 83.2M. While, both GT and GT-Wide do not outperform GIN-VN and DeeperGCN-VN. Especially, we do not observe a performance gain along with the growth of parameters of GT.

Compared to the previous state-of-the-art GNN architecture, Graphormer noticeably surpasses GIN-VN by a large margin, e.g., 11.5% relative validate MAE decline. By using the ensemble with ExpC [51], we got a 0.1200 MAE on complete test set and won the first place of the graph-level track in OGB Large-Scale Challenge[21, 53]. As stated in Section 3.3, we further find that the proposed Graphormer does not encounter the problem of over-smoothing, i.e., the train and validate error keep going down along with the growth of depth and width of models.

## 4.2   Graph Representation

In this section, we further investigate the performance of Graphormer on commonly used graph-level prediction tasks of popular leaderboards, i.e., OGB [22] (OGBG-MolPCBA, OGBG-MolHIV), and benchmarking-GNN [14] (ZINC). Since pre-training is encouraged by OGB, we mainly explore the transferable capability of a Graphormer model pre-trained on OGB-LSC (i.e., PCQM4M-LSC). Please note that the model configurations, hyper-parameters, and the pre-training performance of pre-trained Graphormers used for MolPCBA and MolHIV are different from the models used in the previous subsection. Please refer to Appendix B for detailed descriptions. For benchmarking-GNN, which does not encourage large pre-trained model, we train an additional Graphormer$_{\text{SLIM}}$ ($L = 12, d = 80$, total param.$= 489K$) from scratch on ZINC.

**Baselines.**    We report performance of GNNs which achieve top-performance on the official leaderboards[5] *without additional domain-specific features*. Considering that the pre-trained Graphormer leverages external data, for a fair comparison on OGB datasets, we additionally report performance for fine-tuning GIN-VN pre-trained on PCQM4M-LSC dataset, which achieves the previous state-of-the-art valid and test MAE on that dataset.

**Settings.**    We report detailed training strategies in Appendix B. In addition, Graphormer is more easily trapped in the over-fitting problem due to the large size of the model and the small size of the dataset. Therefore, we employ a widely used data augmentation for graph - FLAG [27], to mitigate the over-fitting problem on OGB datasets.

**Results.**    Table 2, 3 and 4 summarize performance of Graphormer comparing with other GNNs on MolHIV, MolPCBA and ZINC datasets. Especially, GT [13] and SAN [28] in Table 4 are recently proposed Transformer-based GNN models. Graphormer consistently and significantly outperforms previous state-of-the-art GNNs on all three datasets by a large margin. Specially, except Graphormer,

---

[5]https://ogb.stanford.edu/docs/leader_graphprop/
https://github.com/graphdeeplearning/benchmarking-gnns/blob/master/docs/07_
leaderboards.md

Table 2: Results on MolPCBA.

| method | #param. | AP (%) |
|---|---|---|
| DeeperGCN-VN+FLAG [30] | 5.6M | 28.42±0.43 |
| DGN [2] | 6.7M | 28.85±0.30 |
| GINE-VN [5] | 6.1M | 29.17±0.15 |
| PHC-GNN [29] | 1.7M | 29.47±0.26 |
| GINE-APPNP [5] | 6.1M | 29.79±0.30 |
| GIN-VN[50] (fine-tune) | 3.4M | 29.02±0.17 |
| Graphormer-FLAG | 119.5M | **31.39**±0.32 |

Table 3: Results on MolHIV.

| method | #param. | AUC (%) |
|---|---|---|
| GCN-GraphNorm [5, 8] | 526K | 78.83±1.00 |
| PNA [10] | 326K | 79.05±1.32 |
| PHC-GNN [29] | 111K | 79.34±1.16 |
| DeeperGCN-FLAG [30] | 532K | 79.42±1.20 |
| DGN [2] | 114K | 79.70±0.97 |
| GIN-VN[50] (fine-tune) | 3.3M | 77.80±1.82 |
| Graphormer-FLAG | 47.0M | **80.51**±0.53 |

Table 4: Results on ZINC.

| method | #param. | test MAE |
|---|---|---|
| GIN [50] | 509,549 | 0.526±0.051 |
| GraphSage [18] | 505,341 | 0.398±0.002 |
| GAT [47] | 531,345 | 0.384±0.007 |
| GCN [26] | 505,079 | 0.367±0.011 |
| GatedGCN-PE [4] | 505,011 | 0.214±0.006 |
| MPNN (sum) [15] | 480,805 | 0.145±0.007 |
| PNA [10] | 387,155 | 0.142±0.010 |
| GT [13] | 588,929 | 0.226±0.014 |
| SAN [28] | 508,577 | 0.139±0.006 |
| Graphormer$_{SLIM}$ | 489,321 | **0.122**±0.006 |

the other pre-trained GNNs do not achieve competitive performance, which is in line with previous literature [20]. In addition, we conduct more comparisons to fine-tuning the pre-trained GNNs, please refer to Appendix C.

## 4.3 Ablation Studies

We perform a series of ablation studies on the importance of designs in our proposed Graphormer, on PCQM4M-LSC dataset. The ablation results are included in Table 5. To save the computation resources, the Transformer models in table 5 have 12 layers, and are trained for 100K iterations.

**Node Relation Encoding.** We compare previously used positional encoding (PE) to our proposed spatial encoding, which both aim to encode the information of distinct node relation to Transformers. There are various PEs employed by previous Transformer-based GNNs, e.g., Weisfeiler-Lehman-PE (WL-PE) [56] and Laplacian PE [3, 14]. We report the performance for Laplacian PE since it performs well comparing to a series of PEs for Graph Transformer in previous literature [13]. Transformer architecture with the spatial encoding outperforms the counterpart built on the positional encoding, which demonstrates the effectiveness of using spatial encoding to capture the node spatial information.

**Centrality Encoding.** Transformer architecture with degree-based centrality encoding yields a large margin performance boost in comparison to those without centrality information. This indicates that the centrality encoding is indispensable to Transformer architecture for modeling graph data.

**Edge Encoding.** We compare our proposed edge encoding (denoted as via attn bias) to two commonly used edge encodings described in Section 3.1.3 to incorporate edge features into GNN, denoted as via node and via Aggr in Table 5. From the table, the gap of performance is minor between the two conventional methods, but our proposed edge encoding performs significantly better, which indicates that edge encoding as attention bias is more effective for Transformer to capture spatial information on edges.

Table 5: Ablation study results on PCQM4M-LSC dataset with different designs.

| Node Relation Encoding | | Centrality | Edge Encoding | | | valid MAE |
|---|---|---|---|---|---|---|
| Laplacian PE[13] | Spatial | | via node | via Aggr | via attn bias(Eq.7) | |
| - | - | - | - | - | - | 0.2276 |
| ✓ | - | - | - | - | - | 0.1483 |
| - | ✓ | - | - | - | - | 0.1427 |
| - | ✓ | ✓ | - | - | - | 0.1396 |
| - | ✓ | ✓ | ✓ | - | - | 0.1328 |
| - | ✓ | ✓ | - | ✓ | - | 0.1327 |
| - | ✓ | ✓ | - | - | ✓ | 0.1304 |

## 5 Related Work

In this section, we highlight the most recent works which attempt to develop standard Transformer architecture-based GNN or graph structural encoding, but spend less effort on elaborating the works by adapting attention mechanism to GNNs [33, 55, 7, 23, 1, 47, 48, 56, 45].

### 5.1 Graph Transformer

There are several works that study the performance of pure Transformer architectures (stacked by transformer layers) with modifications on graph representation tasks, which are more related to our Graphormer. For example, several parts of the transformer layer are modified in [43], including an additional GNN employed in attention sub-layer to produce vectors of $Q$, $K$, and $V$, long-range residual connection, and two branches of FFN to produce node and edge representations separately. They pre-train their model on 10 million unlabelled molecules and achieve excellent results by fine-tuning on downstream tasks. Attention module is modified to a soft adjacency matrix in [39] by directly adding the adjacency matrix and RDKit[6]-computed inter-atomic distance matrix to the attention probabilites. Very recently, Dwivedi *et al.* [13] revisit a series of works for Transformer-based GNNs, and suggest that the attention mechanism in Transformers on graph data should only aggregate the information from neighborhood (i.e., using adjacent matrix as attention mask) to ensure graph sparsity, and propose to use Laplacian eigenvector as positional encoding. Their model GT surpasses baseline GNNs on graph representation task. A concurrent work [28] propose a novel full Laplacian spectrum to learn the position of each node in a graph, and empirically shows better results than GT.

### 5.2 Structural Encodings in GNNs

**Path and Distance in GNNs.** Information of path and distance is commonly used in GNNs. For example, an attention-based aggregation is proposed in [9] where the node features, edge features, one-hot feature of the distance and ring flag feature are concatenated to calculate the attention probabilites; similar to [9], path-based attention is leveraged in [52] to model the influence between the center node and its higher-order neighbors; a distance-weighted aggregation scheme on graph is proposed in [54]; it has been proved in [32] that adopting distance encoding (i.e., one-hot feature of the distance as extra node attribute) could lead to a strictly more expressive power than the 1-WL test.

**Positional Encoding in Transformer on Graph.** Several works introduce positional encoding (PE) to Transformer-based GNNs to help the model capture the node position information. For example, Graph-BERT [56] introduces three types of PE to embed the node position information to model, i.e., an absolute WL-PE which represents different nodes labeled by Weisfeiler-Lehman algorithm, an intimacy based PE and a hop based PE which are both variant to the sampled subgraphs. Absolute Laplacian PE is employed in [13] and empircal study shows that its performance surpasses the absolute WL-PE used in [56].

**Edge Feature.** Except the conventionally used methods to encode edge feature, which are described in previous section, there are several attempts that exploit how to better encode edge features: an attention-based GNN layer is developed in [16] to encode edge features, where the edge feature

---

[6]https://www.rdkit.org/

is weighted by the similarity of the features of its two nodes; edge feature has been encoded into the popular GIN [50] in [5]; in [13], the authors propose to project edge features to an embedding vector, then multiply it by attention coefficients, and send the result to an additional FFN sub-layer to produce edge representations;

## 6 Conclusion

We have explored the direct application of Transformers to graph representation. With three novel graph structural encodings, the proposed Graphormer works surprisingly well on a wide range of popular benchmark datasets. While these initial results are encouraging, many challenges remain. For example, the quadratic complexity of the self-attention module restricts Graphormer's application on large graphs. Therefore, future development of efficient Graphormer is necessary. Performance improvement could be expected by leveraging domain knowledge-powered encodings on particular graph datasets. Finally, an applicable graph sampling strategy is desired for node representation extraction with Graphormer. We leave them for future works.

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
