# A Proofs

## A.1 SPD can Be Used to Improve WL-Test

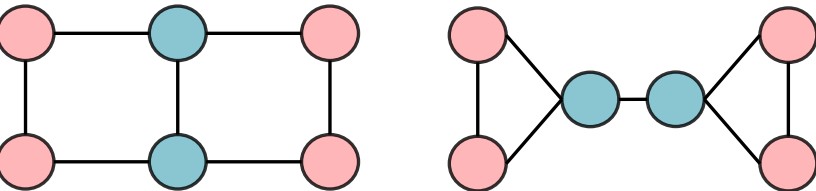

Figure 2: These two graphs cannot be distinguished by 1-WL-test. But the SPD sets, i.e., the SPD from each node to others, are different: The two types of nodes in the left graph have SPD sets $\{0, 1, 1, 2, 2, 3\}$, $\{0, 1, 1, 1, 2, 2\}$ while the nodes in the right graph have SPD sets $\{0, 1, 1, 2, 3, 3\}$, $\{0, 1, 1, 1, 2, 2\}$.

1-WL-test fails in many cases [37, 31], thus classic message passing GNNs also fail to distinguish many pairs of graphs. We show that SPD might help when 1-WL-test fails, for example, in Figure 2 where 1-WL-test fails, the sets of SPD from all nodes to others successfully distinguish the two graphs.

## A.2 Proof of Fact 1

**MEAN AGGREGATE.** We begin by showing that self-attention module with Spatial Encoding can represent MEAN aggregation. This is achieved by in Eq. (6): 1) setting $b_\phi = 0$ if $\phi = 1$ and $b_\phi = -\infty$ otherwise where $\phi$ is the SPD; 2) setting $W_Q = W_K = 0$ and $W_V$ to be the identity matrix. Then $\mathrm{softmax}\,(A)\,V$ gives the average of representations of the neighbors.

**SUM AGGREGATE.** The SUM aggregation can be realized by first perform MEAN aggregation and then multiply the node degrees. Specifically, the node degrees can be extracted from Centrality Encoding by an additional head and be concatenated to the representations after MEAN aggregation. Then the FFN module in Graphormer can represent the function of multiplying the degree to the dimensions of averaged representations by the universal approximation theorem of FFN.

**MAX AGGREGATE.** Representing the MAX aggregation is harder than MEAN and SUM. For each dimension $t$ of the representation vector, we need one head to select the maximal value over $t$-th dimension in the neighbor by in Eq. (6): 1) setting $b_\phi = 0$ if $\phi = 1$ and $b_\phi = -\infty$ otherwise where $\phi$ is the SPD; 2) setting $W_K = e_t$ which is the $t$-th standard basis; $W_Q = 0$ and the bias term (which is ignored in the previous description for simplicity) of $Q$ to be $T\mathbf{1}$; and $W_V = e_t$, where $T$ is the temperature that can be chosen to be large enough so that the softmax function can approximate hard max and $\mathbf{1}$ is the vector whose elements are all 1.

**COMBINE.** The COMBINE step takes the result of AGGREGATE and the previous representation of current node as input. This can be achieved by the AGGREGATE operations described above together with an additional head which outputs the features of present nodes, i.e., in Eq. (6): 1) setting $b_\phi = 0$ if $\phi = 0$ and $b_\phi = -\infty$ otherwise where $\phi$ is the SPD; 2) setting $W_Q = W_K = 0$ and $W_V$ to be the identity matrix. Then the FFN module can approximate any COMBINE function by the universal approximation theorem of FFN.

## A.3 Proof of Fact 2

**MEAN READOUT.** This can be proved by setting $W_Q = W_K = 0$, the bias terms of $Q, K$ to be $T\mathbf{1}$, and $W_V$ to be the identity matrix where $T$ should be much larger than the scale of $b_\phi$ so that $T^2\mathbf{1}\mathbf{1}^\top$ dominates the Spatial Encoding term.

# B Experiment Details

## B.1 Details of Datasets

We summarize the datasets used in this work in Table 6. PCQM4m-LSC is a quantum chemistry graph-level prediction task in recent OGB Large-Scale Challenge, originally curated under the PubChemQC project [40].

Table 6: Statistics of the datasets.

| Dataset | Scale | # Graphs | # Nodes | # Edges | Task Type |
|---------|-------|----------|---------|---------|-----------|
| PCQM4M-LSC | Large | 3,803,453 | 53,814,542 | 55,399,880 | Regression |
| OGBG-MolPCBA | Medium | 437,929 | 11,386,154 | 12,305,805 | Binary classification |
| OGBG-MolHIV | Small | 41,127 | 1,048,738 | 1,130,993 | Binary classification |
| ZINC (sub-set) | Small | 12,000 | 277,920 | 597,960 | Regression |

The task of PCQM4M-LSC is to predict DFT(density functional theory)-calculated HOMO-LUMO energy gap of molecules given their 2D molecular graphs, which is one of the most practically-relevant quantum chemical properties of molecule science. PCQM4M-LSC is unprecedentedly large in scale comparing to other labeled graph-level prediction datasets, which contains more than 3.8M graphs. Besides, we conduct experiments on two molecular graph datasets in popular OGB leaderboards, i.e., OGBG-MolPCBA and OGBG-MolHIV. They are two molecular property prediction datasets with different sizes. The pre-trained knowledge of molecular graph on PCQM4M-LSC could be easily leveraged on these two datasets. We adopt official scaffold split on three datasets following [20, 21]. In addition, we employ another popular leaderboard, i.e., benchmarking-gnn [14]. We use the ZINC datasets, which is the most popular real-world molecular dataset to predict graph property regression for contrained solubility, an important chemical property for designing generative GNNs for molecules. Different from the scaffold spliting in OGB, uniform sampling is adopted in ZINC for data splitting.

## B.2 Details of Training Strategies

### B.2.1 PCQM4M-LSC

Table 7: Model Configurations and Hyper-parameters of Graphormer on PCQM4M-LSC.

| | Graphormer$_{SMALL}$ | Graphormer |
|---|---|---|
| **#Layers** | 6 | 12 |
| **Hidden Dimension** $d$ | 512 | 768 |
| **FFN Inner-layer Dimension** | 512 | 768 |
| **#Attention Heads** | 32 | 32 |
| **Hidden Dimension of Each Head** | 16 | 24 |
| **FFN Dropout** | 0.1 | 0.1 |
| **Attention Dropout** | 0.1 | 0.1 |
| **Embedding Dropout** | 0.0 | 0.0 |
| **Max Steps** | $1M$ | $1M$ |
| **Max Epochs** | 300 | 300 |
| **Peak Learning Rate** | 3e-4 | 2e-4 |
| **Batch Size** | 1024 | 1024 |
| **Warm-up Steps** | $60K$ | $60K$ |
| **Learning Rate Decay** | Linear | Linear |
| **Adam** $\epsilon$ | 1e-8 | 1e-8 |
| **Adam** $(\beta_1, \beta_2)$ | (0.9, 0.999) | (0.9, 0.999) |
| **Gradient Clip Norm** | 5.0 | 5.0 |
| **Weight Decay** | 0.0 | 0.0 |

We report the detailed hyper-parameter settings used for training Graphormer in Table 7. We reduce the FFN inner-layer dimension of $4d$ in [47] to $d$, which does not appreciably hurt the performance but significantly save the parameters. The embedding dropout ratio is set to 0.1 by default in many previous Transformer works [11, 34]. However, we empirically find that a small embedding dropout ratio (e.g., 0.1) would lead to an observable performance drop on validation set of PCQM4M-LSC. One possible reason is that the molecular graph is relative small (i.e., the median of #atoms in each molecule is about 15), making graph property more sensitive to the embeddings of each node. Therefore, we set embedding dropout ratio to 0 on this dataset.

### B.2.2 OGBG-MolPCBA

**Pre-training.** We first report the model configurations and hyper-parameters of the pre-trained Graphormer on PCQM4M-LSC. Empirically, we find that the performance on MolPCBA benefits from the large pre-training model size. Therefore, we train a deep Graphormer with 18 Transformer layers on PCQM4M-LSC. The hidden dimension and FFN inner-layer dimension are set to 1024. We set peak learning rate to 1e-4 for the deep

Table 8: Hyper-parameters for Graphormer on OGBG-MolPCBA, where the **text in bold** denotes the hyper-parameters we eventually use.

|  | Graphormer |
|---|---|
| **Max Epochs** | {2, 5, **10**} |
| **Peak Learning Rate** | {2e-4, **3e-4**} |
| **Batch Size** | 256 |
| **Warm-up Ratio** | 0.06 |
| **Attention Dropout** | 0.3 |
| $m$ | {1, 2,3,**4**} |
| $\alpha$ | 0.001 |
| $\epsilon$ | 0.001 |

Graphormer. Besides, we enlarge the attention dropout ratio from 0.1 to 0.3 in both pre-training and fine-tuning to prevent the model from over-fitting. The rest of hyper-parameters remain unchanged. The pre-trained Graphormer used for MolPCBA achieves a valid MAE of 0.1253 on PCQM4M-LSC, which is slightly worse than the reports in Table 1.

**Fine-tuning.** Table 8 summarizes the hyper-parameters used for fine-tuning Graphormer on OGBG-MolPCBA. We conduct a grid search for several hyper-parameters to find the optimal configuration. The experimental results are reported by the mean of 10 independent runs with random seeds. We use FLAG [26] with minor modifications for graph data augmentation. In particular, except the step size $\alpha$ and the number of steps $m$, we also employ a projection step in [60] with maximum perturbation $\epsilon$. The performance of Graphormer on MolPCBA is quite robust to the hyper-parameters of FLAG. The rest of hyper-parameters are the same with the pre-training model.

### B.2.3  OGBG-MolHIV

Table 9: Hyper-parameters for Graphormer on OGBG-MolHIV, where the **text in bold** denotes the hyper-parameters we eventually use.

|  | Graphormer |
|---|---|
| **Max Epochs** | 8 |
| **Peak Learning Rate** | 2e-4 |
| **Batch Size** | 128 |
| **Warm-up Ratio** | 0.06 |
| **Dropout** | 0.1 |
| **Attention Dropout** | 0.1 |
| $m$ | {1,**2**,3,4} |
| $\alpha$ | {0.001, 0.01, 0.1, **0.2**} |
| $\epsilon$ | {**0**, 0.001, 0.01, 0.1} |

**Pre-training.** We use the Graphormer reported in Table 1 as the pre-trained model for OGBG-MolHIV, where the pre-training hyper-parameters are summarized in Table 7.

**Fine-tuning.** The hyper-parameters for fine-tuning Graphormer on OGBG-MolHIV are presented in Table 9. Empirically, we find that the different choices of hyper-parameters of FLAG (i.e., step size $\alpha$, number of steps $m$, and maximum perturbation $\epsilon$) would greatly affect the performance of Graphormer on OGBG-MolHiv. Therefore, we spend more effort to conduct grid search for hyper-parameters of FLAG. We report the best hyper-parameters by the mean of 10 independent runs with random seeds.

### B.2.4  ZINC

To keep the total parameters of Graphormer less than 500K per the request from benchmarking-GNN leaderboard [14], we train a slim 12-layer Graphormer with hidden dimension of 80, which is called Graphormer$_{\text{SLIM}}$ in Table 4, and has about 489K learnable parameters. The number of attention heads is set to 8. Table 10 summarizes the detailed hyper-parameters on ZINC. We train 400K steps on this dataset, and employ a weight decay of 0.01.

Table 10: Model Configurations and Hyper-parameters on ZINC(sub-set).

| | Graphormer$_{\text{SLIM}}$ |
|---|---|
| **#Layers** | 12 |
| **Hidden Dimension** | 80 |
| **FFN Inner-Layer Hidden Dimension** | 80 |
| **#Attention Heads** | 8 |
| **Hidden Dimension of Each Head** | 10 |
| **FFN Dropout** | 0.1 |
| **Attention Dropout** | 0.1 |
| **Embedding Dropout** | 0.0 |
| **Max Steps** | $400K$ |
| **Max Epochs** | $10K$ |
| **Peak Learning Rate** | 2e-4 |
| **Batch Size** | 256 |
| **Warm-up Steps** | $40K$ |
| **Learning Rate Decay** | Linear |
| **Adam $\epsilon$** | 1e-8 |
| **Adam $(\beta_1, \beta_2)$** | (0.9, 0.999) |
| **Gradient Clip Norm** | 5.0 |
| **Weight Decay** | 0.01 |

Table 11: Hyper-parameters for fine-tuning GROVER on MolHIV and MolPCBA.

| | GROVER | GROVER$_{\text{LARGE}}$ |
|---|---|---|
| **Dropout** | {0.1, 0.5} | {0.1, 0.5} |
| **Max Epochs** | {10, 30, 50} | {10, 30} |
| **Learning Rate** | {5e-5, 1e-4, 5e-4, 1e-3} | {5e-5, 1e-4, 5e-4, 1e-3} |
| **Batch Size** | {64, 128} | {64, 128} |
| **Initial Learning Rate** | 1e-7 | 1e-7 |
| **End Learning Rate** | 1e-9 | 1e-9 |

## B.3 Details of Hyper-parameters for Baseline Methods

In this section, we present the details of our re-implementation of the baseline methods.

### B.3.1 PCQM4M-LSC

The official Github repository of OGB-LSC[6] provides hyper-parameters and codes to reproduce the results on leaderboard. These hyper-parameters work well on almost all popular GNN variants, except the DeeperGCN-VN, which results in a training divergence. Therefore, for DeeperGCN-VN, we follow the official hyper-parameter setting[7] provided by the authors [29]. For a fair comparison to Graphormer, we train a 12-layer DeeperGCN. The hidden dimension is set to 600. The batch size is set to 256. The learning rate is set to 1e-3, and a step learning rate scheduler is employed with the decaying step size and the decaying factor $\gamma$ as 30 epochs and 0.25. The model is trained for 100 epochs.

The default dimension of laplacian PE of GT [13] is set to 8. However, it will cause 2.91% small molecules (less than 8 atoms) to be filtered out. Therefore, for GT and GT-Wide, we set the dimension of laplacian PE to 4, which results in only 0.08% filtering out. We adopt the default hyper-parameter settings described in [13], except that we decrease the learning rate to 1e-4, which leads to a better convergence on PCQM4M-LSC.

### B.3.2 OGBG-MolPCBA

To fine-tune the pre-trained GIN-VN on MolPCBA, we follow the hyper-parameter settings provided in the original OGB paper [21]. To be more concrete, we load the pre-trained checkpoint reported in Table 1 and fine-tune it on OGBG-MolPCBA dataset. We use the grid search on the hyper-parameters for better fine-tuning

---

[6] https://github.com/snap-stanford/ogb/tree/master/examples/lsc/pcqm4m
[7] https://github.com/lightaime/deep_gcns_torch/tree/master/examples/ogb/ogbg_mol#train

Table 12: Comparison to pre-trained Transformer-based GNN on MolHIV. * indicates that additional features for molecule are used.

| method | #param. | AUC (%) |
|---|---|---|
| Morgan Finger Prints + Random Forest* | 230K | **80.60**±0.10 |
| GROVER*[44] | 48.8M | 79.33±0.09 |
| GROVER$_{\text{LARGE}}$*[44] | 107.7M | 80.32±0.14 |
| Graphormer-FLAG | 47.0M | 80.51±0.53 |

Table 13: Comparison to pre-trained Transformer-based GNN on MolPCBA. * indicates that additional features for molecule are used.

| method | #param. | AP (%) |
|---|---|---|
| GROVER*[44] | 48.8M | 16.77±0.36 |
| GROVER$_{\text{LARGE}}$*[44] | 107.7M | 13.05±0.18 |
| Graphormer-FLAG | 47.0M | **31.39**±0.32 |

performance. In particular, the learning rate is selected from $\{1e-5, 1e-4, 1e-3\}$; the dropout ratio is selected from $\{0.0, 0.1, 0.5\}$; the batch size is selected from $\{32, 64\}$.

### B.3.3 OGBG-MolHIV

Similarly, we fine-tune the pre-trained GIN-VN on MolHIV by following the hyper-parameter settings provided in the original OGB paper [21]. We also conduct the grid search to look for optimal hyper-parameters. The ranges for each hyper-parameter of grid search are the same as the previous subsection.

## C  More Experiments

As described in the related work, GROVER is a Transformer-based GNN, which has 100 million parameters and pre-trained on 10 million unlabelled molecules using 250 Nvidia V100 GPUs. In this section, we report the fine-tuning scores of GROVER on MolHIV and MolPCBA, and compare with proposed Graphormer.

We download the pre-trained GROVER models from its official Github webpage[8], follow the official instructions[9] and fine-tune the provided pre-trained checkpoints with careful search of hyper-parameters (in Table 11). We find that GROVER could achieve competitive performance on MolHIV only if employing additional molecular features, i.e., morgan molecular finger prints and 2D features[10]. Therefore, we report the scores of GROVER by taking these two additional molecular features. Please note that, from the leaderboard[11], we can know such additional molecular features are very effective on MolHIV dataset.

Table 12 and 13 summarize the performance of GROVER and GROVER$_{\text{LARGE}}$ comparing with Graphormer on MolHIV and MolPCBA. From the tables, we observe that Graphormer could consistently outperform GROVER even without any additional molecular features.

## D  Discussion & Future Work

**Complexity.**  Similar to regular Transformer, the attention mechanism in Graphormer scales quadratically with the number of nodes $n$ in the input graph, which may be prohibitively expensive for large $n$ and precludes its usage in settings with limited computational resources. Recently, many solutions have been proposed to address this problem in Transformer [24, 50, 55, 36]. This issue would be greatly benefit from the future development of efficient Graphormer.

**Choice of centrality and $\phi$.**  In Graphormer, there are multiple choices for the network centrality and the spatial encoding function $\phi(v_i, v_j)$. For example, one can leverage the $L_2$ distance in 3D structure between two atoms in a molecule. In this paper, we mainly evaluate general centrality and distance metric in graph theory, i.e., the degree centrality and the shortest path. Performance improvement could be expected by leveraging domain knowledge powered encodings on particular graph dataset.

---

[8]`https://github.com/tencent-ailab/grover`
[9]`https://github.com/tencent-ailab/grover/blob/main/README.md#finetuning-with-existing-data`
[10]`https://github.com/tencent-ailab/grover#optional-molecular-feature-extraction-1`
[11]`https://ogb.stanford.edu/docs/leader_graphprop/`

**Node Representation.** There is a wide range of node representation tasks on graph structured data, such as finance, social network, and temporal prediction. Graphormer could be naturally used for node representation extraction with an applicable graph sampling strategy. We leave it for future work.