# OpenReview forum: "Do Transformers Really Perform Badly for Graph Representation?"
_NeurIPS.cc/2021/Conference — NeurIPS 2021 Poster_

### Official Review · Reviewer_8Rq1 · 2021-06-28

**Rating:** 7
**Confidence:** 3

**Summary:**

This paper considers adapting a standard Transformer to work on general graphs, and applying it to a few recent graph representation learning benchmarks. They propose three ways of augmenting a standard Transformer architecture by adding new node features and additive attention biases, and show that their approach (GraphFormer) yields state-of-the-art performance on the considered benchmarks. They also prove some theoretical results stating that GraphFormer is more expressive than many standard graph neural network architectures, and conduct an ablation of the modifications they make to a standard Transformer, showing that each of them leads to better performance.

As the authors note, there has been a lot of work in adapting attention and other parts of transformers to a graph domain. In particular, there has been a lot of work on using attention mechanisms in graphs, and also more specifically in using edges to bias attention between nodes or using features based on shortest paths between nodes. Taken on their own, the modifications proposed in GraphFormer seem fairly incremental.

Nevertheless, most of the prior work I'm aware of focuses on individual modifications for specific tasks instead of a unified set of changes for standardized benchmark datasets, and often deviate from a standard Transformer backbone in a large number of ways. It does seem useful to demonstrate that the Transformer architecture can be applied to graph understanding tasks in a straightforward and general way, and that doing so leads to strong performance both theoretically and empirically. Transformers have shown strong results in many domains, and I think this paper could be a good starting point for future developments on using transformers for graph representation learning.

**Limitations And Societal Impact:**

This work is a methodological contribution without any specific negative societal impacts that I'm aware of.

The authors discuss some limitations of their work, in particular the quadratic complexity of global self attention and the need for future work on graph subsampling strategies. They also mention that their approach may be more prone to overfitting, although regularization seems to fix this. (The checklist mentions section 5 containing limitations, but I think the authors may have meant section 6?)

**Main Review:**

## Proposed approach
The authors propose three ways of adapting a standard Transformer architecture to work on graphs.

- First, they propose adding in-degree and out-degree embeddings to the embedding of each node, before computing the query and key attention vectors. This gives the network an ability to condition on those degrees in addition to the local features of the node itself.
- Second, they propose computing the distance between each pair of nodes, and then using this to choose from a set of learnable biases for the attention logit between that pair of nodes. In particular, by assigning different biases, the network can learn to focus attention only on nodes in the neighborhood, or to nodes far away, with different weights assigned to each. This adjustment happens "outside" the key-value dot product, so it is based on distance alone and not based on the query and key embeddings.
- Third, for graphs with edge features, they propose adding an adjustment based on the ordered sequence of edges along the shortest path between those nodes. In particular, for each possible index $n$ in the shortest path, they learn an edge weight vector ${{w^E}\_{n}}$, then take the dot product between ${{w^E}\_{n}}$ and the $n$-th edge feature ${{x}\_{e\_n}}$ along the shortest path. This is also added to the attention logits directly, allowing each attention head to learn to attend to a different type of path between nodes.

In addition to these modifications, they adapt some well-known tricks from transformers and graph neural networks:
- They apply layer normalization before attention, which has been shown to work well for transformers in other domains,
- They introduce a "special" node [VNODE] that tracks full-graph information, similar to the use of similar nodes in MPNNs and the use of the [CLS] token in BERT. However, they emphasize that this node is "virtual" and doesn't participate in the shortest-path embeddings (it instead gets its own, separate attention bias).

One question I had about the edge feature path embedding: what do you do when there are multiple paths of the same length? Should all of them be averaged together, or do you just pick one? (Does it matter?)

## Theoretical and empirical results
Theoretically, the authors argue that GraphFormer is able to represent the same operations as a standard GNN: in particular, that it can mask attention to the nearest neighbors, use the known degree to compute sums over neighbors, and use the FFN in the attention layer to combine node and neighbor features. Their proof defers to the universal approximation theorem of fully-connected networks, and although it is a fairly informal argument I think it's essentially correct. (I think it would be a bit more precise to say that GraphFormer can approximate any other GNN arbitrarily well with enough parameters, since that's all the universal approximation theorem allows?)

They also argue that GraphFormer is not limited by the 1-WL test, by providing an example where distances between pairs of nodes can be used to distinguish two graphs that the 1-WL test cannot distinguish. Finally, they argue that even without a special [VNODE] node, GraphFormer can still learn to aggregate information across the graph using the "mean readout" function (essentially, by having all of the queries, keys, and spatial biases be identical so that every node attends to every other node).

Empirically, they demonstrate their approach on the OGB Large-Scale Challenge, OGB, and benchmarking-GNN benchmarks. For each, they compare to a variety of graph neural network approaches that have been previously applied to those benchmarks. They also compare to the Graph Transformer model of Dwivedi and Bresson [13], a different general-purpose adaptation of transformers to graphs. The GraphFormer model shows strong results across the different domains, outperforming all of the reported baselines.

The authors also include an ablation study on one of the datasets, showing that each of their proposed modifications improves performance over a vanilla Transformer, as well as over some alternative encoding strategies from prior work.

One thing that would strengthen the experimental results would be to compare the GraphFormer architecture to some of the other methods for encoding paths, distances, and edge features that the authors cite in their related work section. For instance, how does GraphFormer compare to the concatenated-path-attention strategy in [9] or the distance encoding of [30]?

## Originality and significance
As the authors note, the success of transformer-based models has led to a wide variety of work on combining parts of graph neural networks with parts of transformers. Thus, many of the ideas used in GraphFormer (in particular, using attention mechanisms locally and globally, and biasing attention using shortest paths, edge features, or distances between nodes) resemble things from prior work (and the authors do a good job of citing many of these earlier works).

Even so, one of the nice things about the proposed GraphFormer architecture is that the modifications to the standard transformer architecture are quite simple and general. The node degree information is simply added as part of the node features, and the distance and path features are added as attention biases. This suggests that improvements to transformers for NLP or other domains could easily transfer to improvements in GraphFormer, and vice versa.

Thus, despite the simplicity of the proposed changes, I think this paper could be quite impactful by showing that, both theoretically and empirically, the standard Transformer architecture can be successfully adapted to graph representation learning tasks. In fact, the simplicity might be a good thing, as it provides a signal that the Transformer backbone can do very well with just a few extra graph-based features. I wouldn't be surprised if this was a starting point for additional research on applying transformers to graphs, even if not all of those future directions incorporate exactly the same set of modifications.

One piece of related work that the authors may be unaware of is ["Global relational models of source code" (Hellendoorn et al. 2019)](https://openreview.net/forum?id=B1lnbRNtwr), which proposes adding edge-feature-based attention biases to a standard transformer, applied to a graph representation of a program.

## Clarity
Overall I found the paper easy to read and well motivated. The authors provide an intuitive explanation for why each of their modifications is sensible, and give a nice proof sketch in the main text of how the GraphFormer architecture generalizes other GNN architectures.

Minor suggestions for clarification:
- Line 253: "transferable capability of the pre-trained GraphFormer on OGB-LSC": I initially interpreted this as transferring from something else to OGB-LSC. But I think you actually mean transferring from OGB-LSC to graph classification? Perhaps you could reword this to "transferable capability of a GraphFormer model pre-trained on OGB-LSC" or something similar?
- Line 262: Is PCQM4M-LSC the same dataset you pretrain GraphFormer on, or did you pretrain on a different dataset?


**Time Spent Reviewing:**

2.5

---

> ### Author Response · Authors · 2021-08-09
> **Response To Reviewer 8Rq1**
>
> We sincerely thank you very much for appreciating our work! We are deeply grateful for your extremely careful review and invaluable suggestions! We address your concerns as follow:
>
> **Q1: What do you do when there are multiple shortest paths (SP)? Does it matter?**
>
> A: In our work, we randomly pick one if there are multiple SPs. To better understand whether the sampling is a big issue, we sampled a subset of graphs in the PCQM4M dataset (10K molecular graphs) and counted the number of SPs between all node pairs for each graph. We find that 86.98% of node pairs have only one SP, 12.25% have two SPs, and less than 1% have more than two SPs. Interestingly, we find that for most node pairs with two SPs, the two SPs are symmetric, e.g., two carbon atoms opposite each other on the benzene ring. So randomly picking any SP would give the same representation and never hurt the model training/inference. We will make further investigation on graphs with diverse types of data beyond the molecule.
>
> **Q2: It would be a bit more precise to say that GraphFormer can approximate any other GNN arbitrarily well with enough parameters.**
>
> A: Thanks very much for the suggestions. We will follow your advice in the next version.
>
> **Q3: Comparing to PAGTN and Distance Encoding.**
>
> A: We conduct experiments on QM7, which is a dataset of quantum mechanics, and compare GraphFormer to PAGTN, distance encoding (DE-GCN) and other baseline GNNs.
>
> | Task  | Metric    | GAT[1]  | GIN[2] | GCNN[3] | GCN[4] |  N-Gram Graph[5]  |  PAGTN[6] | DE-GCN[7]|  GraphFormer  |
> | ----- | --------- | ------- | ------ | ------- | ------ | ----------------- | --------- | -------- | ------------- |
> | QM7   | Valid MAE | 213.0   | 82.7   | 76.0    | 52.4   |     49.7          |  47.8     |   43.4   |   14.87       |
>
> As we can see in the table, GraphFormer could outperform baseline methods by a large margin.
>
> **Q4: Related work of ''Global relational models of source code''.**
>
> A: Thanks very much for pointing out a related work of our paper. We will discuss it in the next version.
>
> **Q5: ''transferable capability of the pre-trained GraphFormer on OGB-LSC''.**
>
> A: Sorry for the misleading. Yes, your understanding is right. It should be ''transferable capability of a GraphFormer model pre-trained on OGB-LSC'', and we use PCQM4M-LSC as the pre-training dataset. We will clarify our writing accordingly in the next version.
>
> We appreciate the reviewer for spending time to review our paper and offer constructive suggestions. We truly hope that our response could address your concerns.
>
> [1] P. Velicˇkovic´, G. Cucurull, A. Casanova, A. Romero, P. Lio, and Y. Bengio. Graph attention networks. ICLR 2018.
>
> [2] K. Xu, W. Hu, J. Leskovec, and S. Jegelka. How powerful are graph neural networks? ICLR 2019.
>
> [3] D. K. Duvenaud, D. Maclaurin, J. Iparraguirre, R. Bombarell, T. Hirzel, A. Aspuru-Guzik, and R. P. Adams. Convolutional networks on graphs for learning molecular fingerprints. NIPS, 2015.
>
> [4] Kipf T N, Welling M. Semi-supervised classification with graph convolutional networks[J]. ICLR 2017.
>
> [5] S. Liu, M. F. Demirel, and Y. Liang. N-gram graph: Simple unsupervised representation for graphs, with applications to molecules. NIPS, 2019.
>
> [6] Chen B, Barzilay R, Jaakkola T. Path-augmented graph transformer network[J]. arXiv preprint arXiv:1905.12712, 2019.
>
> [7] Li P, Wang Y, Wang H, et al. Distance Encoding: Design Provably More Powerful Neural Networks for Graph Representation Learning[J]. NeurIPS, 2020.

---

> > ### Comment · Reviewer_8Rq1 · 2021-08-13
> > **Response to author clarifications**
> >
> > Thank you for the clarifications. Your response addresses my concerns, and the additional results are impressive.
> >
> > I do wonder if it would be better to average over all shortest paths, instead of randomly sampling one. But for the PCQM4M dataset, at least, it sounds like it would not matter most of the time. In either case, it would be good to add a brief discussion of this choice in the paper.

---

> > > ### Author Response · Authors · 2021-08-16
> > > **Thank you**
> > >
> > > Thanks very much for the valuable advice. We will carefully make discussions in the next version.

---

### Official Review · Reviewer_uohB · 2021-07-12

**Rating:** 6
**Confidence:** 3

**Summary:**

This work proposes a transformer architecture suited for graph representation. More precisely, the structure of the graph is encoded via three different mechanisms focused on the degrees, the distance of the shortest path between two nodes, and the edges. The resulting model outperforms different GNNs on four molecular graph tasks.

**Ethical Concerns:**

None.

**Limitations And Societal Impact:**

The authors identify sensible limitations in their work. Negative societal impact are not mentioned, although they remain limited.

**Main Review:**

As GNNs are the current preferred architecture for graph representations, outperforming these make this work significant. The paper is also sound and clear. Some more experiments may be useful to better understand the capabilities of GraphFormer as well as some clarifications on the novelty of the encoding mechanisms.

Pros:
- The proposed architecture seems sound and simple.
- GraphFormer obtains excellent results on OGB.

Cons:
- The paper claims that the proposed Spatial Encoding is novel (l129). In my opinion, this may be exaggerated as this encoding seems rather be an instance of relative position encoding, where a parameter depending on the distance (of the shortest path) between two nodes are learned.
- The paper does not propose experiments on smaller molecular data sets (a few hundreds/thousands graphs). This could be useful to understand how does GraphFormer do with respect to GNNs in different settings. To my understanding, these experiments could be conducted with and without pre-training.

Questions and remarks:
- l18: what do you mean by "powerful"?
- Experiments: GAT, which also applies attention (although on neighboring nodes only) could be an interesting baseline for this work, why not comparing to it?

**Time Spent Reviewing:**

4

---

> ### Author Response · Authors · 2021-08-09
> **Response To Reviewer uohB**
>
> We sincerely thank you for reviewing our paper and providing valuable suggestions. We address your concerns as follows:
>
> **Q1: The novelty of our proposed spatial encoding with respect to the relative positional encoding in NLP.**
>
> A: This is a good catch. Our spatial encoding is related to relative positional encoding in sequence modeling, as the sequence is indeed a special kind of graph where all nodes are arranged in a line. See discussions in lines 121-127. However, the relationship between two nodes in a graph can be more complicated than that between two positions in a line. For example, sometimes there exists two nodes that are not connected, which doesn't not happen in natural sentences. We carefully take all the possible situations into the spatial encoding, which actually generalizes relative positional encoding in sequence modeling.
>
> **Q2: Experiments on smaller molecular datasets (a few hundreds/thousands graphs).**
>
> A: In our paper, we conduct experiments on the ZINC dataset, which contains only 11K molecular graphs. To address your concern, we further conduct an experiment on the QM7 dataset, a quantum mechanics dataset containing only 7K graphs. The architecture and training strategies of GraphFormer used for this task are exactly the same as those used in the ZINC experiment, except that we only train 200 epochs.
>
> We randomly split the samples for train/val/test by 8:1:1, and compare our method to baseline GNNs which achieve top-performance on this task.
>
> | Task  | Metric    | GAT[1]  | GIN[2] | GCNN[3] | GCN[4] |  N-Gram Graph[5]  |  PAGTN[6] |  GraphFormer  |
> | ----- | --------- | ------- | ------ | ------- | ------ | ----------------- | --------- | ------------- |
> | QM7   | Valid MAE | 213.0   | 82.7   | 76.0    | 52.4   |     49.7          |  47.8     |   14.87       |
>
> As we can see in the table, GraphFormer still outperform baseline GNNs by a large margin on the smaller dataset.
>
>
> **Q3: What do you mean by ''powerful''?**
>
> A: Thanks for this question. We say Transformer is powerful for its great expressiveness and adaptability. Previous works [7,8] prove that Transformer model is a universal approximator of sequence-to-sequence functions, and some Transformer variants are even Turing-complete. We will make a more precise expression in the next version of the paper.
>
> **Q4: Why not compare to GAT?**
>
> When comparing GraphFormer and previous works, we choose the top-performance GNNs on official leaderboards as our baseline methods. However, we found that GAT does not perform well on many graph representation learning tasks, and thus we didn't include it. To address your concern, we report the performance of GAT on the QM7 dataset in the table above. It can be clearly seen that GAT performs much worse than all the baselines and GraphFormer.
>
> We appreciate the reviewer for spending time to review our paper and offer constructive suggestions. We hope that our response could address your concerns. If you are satisfied by the response, please kindly reconsider your score.
>
>
> [1] P. Velicˇkovic´, G. Cucurull, A. Casanova, A. Romero, P. Lio, and Y. Bengio. Graph attention networks. ICLR 2018.
>
> [2] K. Xu, W. Hu, J. Leskovec, and S. Jegelka. How powerful are graph neural networks? ICLR 2019.
>
> [3] D. K. Duvenaud, D. Maclaurin, J. Iparraguirre, R. Bombarell, T. Hirzel, A. Aspuru-Guzik, and R. P. Adams. Convolutional networks on graphs for learning molecular fingerprints. NIPS, 2015.
>
> [4] Kipf T N, Welling M. Semi-supervised classification with graph convolutional networks[J]. ICLR 2017.
>
> [5] S. Liu, M. F. Demirel, and Y. Liang. N-gram graph: Simple unsupervised representation for graphs, with applications to molecules. NIPS, 2019.
>
> [6] Chen B, Barzilay R, Jaakkola T. Path-augmented graph transformer network[J]. arXiv preprint arXiv:1905.12712, 2019.
>
> [7] Yun C, Bhojanapalli S, Rawat A S, et al. Are Transformers universal approximators of sequence-to-sequence functions?[C] ICLR. 2019.
>
> [8] Dehghani M, Gouws S, Vinyals O, et al. Universal Transformers[C]//ICLR. 2018.

---

> > ### Comment · Reviewer_uohB · 2021-09-01
> > **Noted**
> >
> > Thank you for your comment which clarifies my concerns on relative positional encoding and GAT. I still think that it would have been interesting to see how your model fares in more data constrained settings than QM7 but I do not blame you for this as my answer arrives late

---

> > > ### Author Response · Authors · 2021-09-02
> > > **Thank you**
> > >
> > > Thank you for the valuable comments.
> > >
> > > We follow your suggestion "...experiments on smaller molecular data sets (a few hundreds/thousands graphs)..." to conduct experiment on QM7 dataset which only contains 7165 molecule samples. Could you please kindly name a few "data constrained" datasets so that we would  report the results in next revision, if we have enough time.
> > >
> > > We appreciate the reviewer for spending time to review our paper and offer constructive suggestions. If you are satisfied by the response, would you please kindly reconsider your score?

---

### Official Review · Reviewer_3WSK · 2021-07-16

**Rating:** 7
**Confidence:** 4

**Summary:**

GraphFormer modifies the attention layer in a Transformer in a way to incorporate information about the structure of the graph. The modifications are intuitively motivated and show that GraphFormer is at least as powerful as GNNs. The experimental results are positive although they seem to come at the cost of many more parameters.

**Limitations And Societal Impact:**

I don't see any potential negative societal impact.

The authors state that performance, especially on large graphs, will limit applicability of their system.

**Main Review:**

Originality: The idea appears to motivated by Graph Transformer and several other papers that add structural encodings to GNNs as mentioned in the Related Work section. Adding structural encodings to the attention mechanism indeed appears novel.

Quality: Simple proofs are provided to show that the model is at least as powerful as GNNs. Experiments are run on Open Graph Benchmark (OGB) which is standard for deep learning on graphs. Results are strong. Ablations are done to show the importance of each structural encoding ingredient.

Clarity: Background knowledge is clear and diagrams are good. Information is provided to reproduce experiments.

Significance: The method is easy to apply to any graph, and the experimental results are strong. Given the familiarity of Transformers, I expect many researchers and partitioners will reach for this model when presented with graph data. The main limitation appears to be the quadratic scaling with the size of the graph.

**Time Spent Reviewing:**

2

---

> ### Author Response · Authors · 2021-08-09
> **Response To Reviewer 3WSK**
>
> We sincerely thank you for reviewing our paper and appreciating our work.
>
> **For the concerns about the model scale**, in Table 1, we can see that GraphFormer_{small} has only 12.5M parameters, which is less than many baseline models (GINE-vn(13.2M), DeeperGCN-vn(25.5M), and GT-wide(83.2M)), but significantly outperforms the baselines by a large margin. In Table 4, the number of parameters in GraphFormer_{slim} is less than most of baseline models, but it achieves the best test MAE on the ZINC leaderboard. This clearly demonstrates that Graphormer has great strength to capture the graph representation than the baseline models even with a similar model size.
>
> We kindly hope that our response could address your concerns.

---

### Official Review · Reviewer_rR8v · 2021-07-17

**Rating:** 7
**Confidence:** 5

**Summary:**

The paper proposed a simple Transformer-based neural network architecture for graph data, called GraphFormer. Inspired by the positional encoding in NLP, GraphFormer encodes the structural information of the graph into the Transformer architecture via 1) centrality encoding, which incorporates degree information in computing the node embeddings, 2) spatial and edge encoding, which utilizes the shortest path between nodes to determine the edge biases in self-attention. In addition, the author added a set of virtual nodes to GraphFormer that connects to all nodes via "virtual" edges. The author compared GraphFormer with other models in several molecule/chemistry graph classification tasks and show that GraphFormer significantly outperforms the other methods, including several SOTA Message-Passing Graph Neural Network (MPGNN) models.


**Ethical Concerns:**

No concern.

**Limitations And Societal Impact:**

I think the author needs to state that GraphFormer may not be suitable for large graphs due to the full-attention. In addition, the author needs to emphasize on the more extensive computational cost of the model.


**Main Review:**


- Originality
The architecture of the paper is similar to the Code Transformer model proposed in [ICLR2021, Language-Agnostic Representation Learning of Source Code from Structure and Context]. However, the author has not mentioned this paper in the related work. Other than that, the usage of virtual nodes and the way of generating the edge encodings via shortest path are novel and original.

- Technical Quality
The author performed extensive experimental evaluations of GraphFormer on benchmarks like OGB and ZINC and demonstrated that it achieves state-of-the-art performance. However, from Table 1,2,3, we can notice that GraphFormer has significantly more parameters than the baselines. This means that GraphFormer can be much more computationally expensive than the other models. Thus, it is not clear if the performance improvement of GraphFormer is attributed to its larger model size or architectural advances. In addition, one limitation of the encoding mechanism adopted by GraphFormer is that it may not be efficient for graphs with "hubs" (nodes with very large degrees). The author may need to mention this limitation in the paper.

- Clarity
The paper is well-written and easy to understand.

- Significance
Performance of GraphFormer is significantly better than the baselines according to Table 1,2,3. However, I need to see more evidence to understand if it is simply attributed to scaling up the model size or due to the usage of Transformer.


--- Post Rebuttal ---
The author has address my concerns and I will increase the score to accept.

**Time Spent Reviewing:**

24

---

> ### Author Response · Authors · 2021-08-09
> **Response To Reviewer rR8v**
>
> We sincerely thank the reviewer for reviewing our paper and providing valuable suggestions. We address your concerns as follows:
>
> **Q1. Related work of ''Code Transformer''.**
>
> A: Thank you very much for providing a reference for our paper. It is relevant to our work, and we will carefully make discussions in the related work section.
>
> **Q2. Whether the performance gain of GraphFormer comes from its larger model size?**
>
> A: Thanks for the question. The performance gain cannot be considered as entirely coming from its larger model size. In Table 1, GraphFormer_{small} has only 12.5M parameters, which is less than many baseline models (GINE-vn(13.2M), DeeperGCN-vn(25.5M), and GT-wide(83.2M)), but significantly outperforms the baselines by a large margin. In Table 4, the number of parameters in GraphFormer_{slim} is less than most of baseline models but it achieves the best test MAE on the ZINC leaderboard.
>
> This clearly demonstrates that Graphormer has great strength to capture the graph representation effectively. Increasing parameters of GraphFormer can further boost the model performance (see GraphFormer_{small} v.s. GraphFormer in Table 1).
>
>
> **Q3: The computational complexity of ''hubs'' (nodes with large degrees) in graph is costly.**
>
> A: The computational complexity of the model mainly depends on the number of nodes in the graph but does not depend on the node degree. In equation 7, it can be seen that the complexity of the self-attention is O(n^2), where n=|V| is the number of nodes. For the centrality encoding, we give each node a real-valued vector depending on its degree, which can be done in the preprocessing stage with one pass.
>
>
> **Q4: The full attention mechanism used in GraphFormer will introduce heavy computational cost for large graph.**
>
> A: It is a good catch. We discussed the computational complexity problem of GraphFormer in the conclusion section and will move it to the main body of the paper in the next version.
>
> Note that the Transformer model has many efficient approximation methods, which leads to a sub-quadratic complexity. See a comprehensive survey paper[1]. GraphFormer can naturally integrate with those modifications, making it scalable to large graphs. We would like to leave it for future work.
>
> We appreciate the reviewer for spending time to review our paper and offer constructive suggestions. We hope that our response could address your concerns. If you are satisfied with our response, please kindly reconsider your score.
>
> [1] Tay Y, Dehghani M, Bahri D, et al. Efficient transformers: A survey[J]. arXiv preprint arXiv:2009.06732, 2020.

---

### Decision · Program_Chairs · 2021-09-27

**Decision:**

Accept (Poster)

**Comment:**

This paper presents a set of simple modifications to the popular transformer architecture so that they can work well on graph structured data.  The paper is quite clear and well-written.  Extensive empirical results show that these architectures can outperform graph neural networks on a wide range of benchmarks.

All reviewers consistently recommend accepting this paper, and are optimistic about the adoption of the proposed techniques due to their simplicity and strong empirical results.

The authors should consider the reviewers’ suggestions to improve the paper, in particular the additional related works mentioned in the reviews that can better help put this work into context.